# Light-induced stomatal opening requires phosphorylation of the C-terminal auto-inhibitory domain of plasma membrane H$^+$-ATPase

Saashia Fuji [1,7], Shota Yamauchi[1,6,7], Naoyuki Sugiyama[2], Takayuki Kohchi [3], Ryuichi Nishihama [3,4], Ken-ichiro Shimazaki[5] & Atsushi Takemiya [1] ✉

Plasma membrane H$^+$-ATPase provides the driving force for light-induced stomatal opening. However, the mechanisms underlying the regulation of its activity remain unclear. Here, we show that the phosphorylation of two Thr residues in the C-terminal autoinhibitory domain is crucial for H$^+$-ATPase activation and stomatal opening in *Arabidopsis thaliana*. Using phosphoproteome analysis, we show that blue light induces the phosphorylation of Thr-881 within the C-terminal region I, in addition to penultimate Thr-948 in AUTOINHIBITED H$^+$-ATPASE 1 (AHA1). Based on site-directed mutagenesis experiments, phosphorylation of both Thr residues is essential for H$^+$ pumping and stomatal opening in response to blue light. Thr-948 phosphorylation is a prerequisite for Thr-881 phosphorylation by blue light. Additionally, red light-driven guard cell photosynthesis induces Thr-881 phosphorylation, possibly contributing to red light-dependent stomatal opening. Our findings provide mechanistic insights into H$^+$-ATPase activation that exploits the ion transport across the plasma membrane and light signalling network in guard cells.

Stomata, i.e., tiny pores on the surface of leaves in land plants, play a vital role in plant development as well as in the regulation of gas exchange between plants and the atmosphere[1–5]. Stomata open in response to light, facilitating the uptake of $CO_2$ for photosynthetic carbon fixation and delivery of mineral nutrients absorbed by roots from the soil via transpiration. Light-induced stomatal opening consists of two distinct mechanisms, namely red and blue light responses, and these stimuli increase turgor pressure within a pair of guard cells, thus inducing stomatal opening[6–8]. The red light response relies on photosynthesis and requires a high intensity of red light, whereas the

blue light-specific response can be induced by a low intensity of blue light triggered by receptor kinase phototropins. The blue light response is enhanced depending on the background red light intensity, and the stomatal aperture illuminated simultaneously with blue and red light is larger than the sum of the apertures illuminated with blue and red light alone[6,8]. Such a synergistic effect of blue and red light on stomatal opening is partially explained by the interaction of phototropin-mediated responses with responses caused by the reduction in intercellular $CO_2$ concentration ($Ci$) in leaves through photosynthetic $CO_2$ assimilation[9–12].

[1]Department of Biology, Graduate School of Sciences and Technology for Innovation, Yamaguchi University, 1677-1 Yoshida, Yamaguchi 753-8512, Japan. [2]Department of Molecular & Cellular BioAnalysis, Graduate School of Pharmaceutical Sciences, Kyoto University, Kyoto 606-8501, Japan. [3]Graduate School of Biostudies, Kyoto University, Kyoto 606-8502, Japan. [4]Department of Applied Biological Science, Faculty of Science and Technology, Tokyo University of Science, 2641 Yamazaki, Noda, Chiba 278-8510, Japan. [5]Department of Biology, Faculty of Science, Kyushu University, 744 Motooka, Fukuoka 819-0395, Japan. [6]Present address: Department of Applied Biological Science, Faculty of Science and Technology, Tokyo University of Science, Chiba, Japan. [7]These authors contributed equally: Saashia Fuji, Shota Yamauchi. ✉e-mail: take.pcs@yamaguchi-u.ac.jp

Phototropins (phot1 and phot2) are plant-specific blue light receptors comprising two photosensory light, oxygen, or voltage (LOV) domains at the N-terminus and a Ser/Thr kinase domain at the C-terminus[13,14]. Blue light perception by LOV domains activates their kinase domain and undergoes autophosphorylation at the Ser residues in the kinase activation loop, which is a prerequisite for phototropin-mediated responses, including stomatal opening, phototropism, chloroplast movements, and leaf expansion[14,15]. Phototropins phosphorylate their downstream substrates to convert and transmit light signals into intracellular signals, thereby inducing various physiological responses. Genetic and biochemical studies have identified BLUE LIGHT SIGNALING1 (BLUS1), a guard cell-specific Ser/Thr protein kinase essential for blue light-dependent stomatal opening[16]. Phototropins physically interact with BLUS1 and phosphorylate Ser-348 in the C-terminal regulatory domain of BLUS1 in a blue light-dependent manner, which abates autoinhibition of the kinase domain[12,16,17]. Activated BLUS1 transduces signals through downstream components, such as the group C1 Raf-like MAP kinase kinase kinase (MAPKKK) BLUE LIGHT-DEPENDENT H⁺-ATPASE PHOSPHORYLATION (BHP) and type 1 protein phosphatase (PP1), ultimately activating plasma membrane H⁺-ATPase[18–20]. Phototropins also phosphorylate another substrate, namely the CONVERGENCE OF BLUE LIGHT AND $CO_2$ 1 (CBC1), which is a group C7 Raf-like MAPKKK[21]. CBC1, together with its paralogue CBC2, suppresses plasma membrane S-type anion channels via blue light[21,22]. Both blue light-dependent H⁺-ATPase activation and anion channel inhibition contribute to plasma membrane hyperpolarisation, which drives K⁺ uptake via voltage-gated inward-rectifying K⁺ channels[23]. K⁺ accumulation in the guard cells leads to water influx, resulting in increased turgor pressure and stomatal opening.

Plasma membrane H⁺-ATPase is an electrogenic pump highly conserved among plants, algae, and fungi[24]. It exports cellular H⁺ coupled with ATP hydrolysis, creating an electrochemical H⁺ gradient across the plasma membrane, which energises the fundamental transport processes[25]. In the *Arabidopsis thaliana* genome, 11 genes encoding functional H⁺-ATPase, designated *AUTOINHIBITED H⁺-ATPASE 1−11* (*AHA1–AHA11*), have been identified. Owing to its physiological significance, plant H⁺-ATPase is expressed in various cell types, such as guard cells, phloem companion cells, bundle-sheath cells, pollen cells, and root epidermis[26–28]. The plasma membrane H⁺-ATPase has 10 transmembrane segments with cytoplasmic actuator (A), nucleotide-binding (N), and phosphorylation (P) domains responsible for catalytic activity[29]. The C-terminal peptide, composed of approximately 100 residues, functions as an autoinhibitory domain and inhibits enzyme activity by interfering with the catalytic domain[30–32]. Blue light induces the phosphorylation of the penultimate Thr residue of H⁺-ATPase and subsequent binding of 14-3-3 proteins in guard cells[33]. Numerous studies that used a system heterologously expressing plant H⁺-ATPase in a yeast strain conditionally devoid of endogenous plasma membrane H⁺-ATPase *PMA1* have suggested that penultimate Thr phosphorylation and phosphorylation-dependent 14-3-3 protein binding may reverse C-terminal autoinhibition and activate H⁺-ATPase[34]. However, genetic evidence regarding whether penultimate Thr phosphorylation is involved in H⁺-ATPase activation in plants is lacking.

Furthermore, recent studies have shown that, in addition to the penultimate Thr, phosphorylation of the Thr residue upstream of the C-terminal autoinhibitory domain causes H⁺-ATPase activation. The inhibitory effect of the C-terminal domain on H⁺-ATPase activity is thought to be caused by two regions (regions I and II) which possibly interact and interfere with the catalytic domains[32,35]. The Thr-881 of AHA1 in region I is highly conserved in plants, and a recent study showed that AHA2 phosphorylation at Thr-881 is enhanced in *Arabidopsis* roots under low K⁺ conditions[36]. Mutational experiments using a yeast heterologous expression system have indicated that the phosphorylation of Thr-881 increases H⁺-ATPase activity without affecting

14-3-3 protein binding[37]. However, it is unclear whether Thr-881 is also phosphorylated by blue light in guard cells, and the functional significance of Thr-881 phosphorylation in plants has yet to be verified.

A previous thermal imaging analysis identified an *Arabidopsis aha1* mutant defective in stomatal opening in response to blue light[38]. The *aha1* mutant reduced the protein expression of total H⁺-ATPase in guard cells to 30% that of wild-type plants, suggesting that AHA1 is the major isoform of the plasma membrane H⁺-ATPase in guard cells[38]. Importantly, blue light-dependent H⁺ pumping in guard cell protoplasts was significantly attenuated in the *aha1* mutant, and the expression of wild-type *AHA1* gene in the *aha1* mutant under the control of the native *AHA1* promoter restored the stomatal blue light responses[38]. Thus, guard cells from the *aha1* mutant offer a reliable system for the functional analysis of H⁺-ATPase activation process in plants.

In the present study, we performed a phosphoproteome analysis using guard cell protoplasts and found that the Thr-881 of AHA1, together with the penultimate Thr-948, was phosphorylated in response to blue light. We showed that the phosphorylation of both Thr-881 and Thr-948 is crucial for H⁺-ATPase activation, which allows stomatal opening. Furthermore, we found that Thr-881 was phosphorylated by guard cell photosynthesis, which may contribute to the mechanism of red light-dependent stomatal opening in intact leaves.

## Results

### Blue light induces phosphorylation of AHA1 at Thr-881 and Thr-948 in the guard cells

The guard cell protoplasts from the wild type and *phot1-5 phot2-1* double mutant were illuminated with high-fluence-rate red light for 30 min, followed by a short pulse of blue light for 30 s superimposed on a background red light[16,21]. Trichloroacetic acid was added to the protoplasts to recover the proteins before and after blue-light irradiation, and trypsin and Lys-C were used to cleave them into peptides. The phosphopeptides were enriched from the digested samples by hydroxy acid-modified metal oxide chromatography (HAMMOC)[39] and analysed by nano-liquid chromatography tandem mass spectrometry (nanoLC/MS/MS). Phosphopeptides that were phosphorylated in response to blue light in a phototropin-dependent manner, including the phototropin substrates BLUS1 and CBC1, were identified[16,21]. In these analyses, blue light-dependent phosphorylation of the penultimate Thr residue of AHA1 (Thr-948) was also identified (Fig. 1a, b; ref. 21.). Furthermore, the phosphorylation level of AHA1 at Thr-881, located in region I within the C-terminal autoinhibitory domain, was also elevated in response to blue light (Fig. 1a, b). In contrast, blue light-dependent phosphorylation of both Thr-881 and Thr-948 was not detected in the *phot1-5 phot2-1* mutant (Fig. 1b). Antibodies recognising phosphorylated Thr-881 and Thr-948 of AHA1 were produced, and it was confirmed that both Thr residues were phosphorylated in response to blue light in the guard cells of the wild-type but not in those of the *phot1-5 phot2-1* mutant (Fig. 1c, d). Both Thr-881 and Thr-948 were conserved among the 11 AHA isoforms, except for AHA10, in which the amino acid residue corresponding to Thr-881 was replaced by a hydroxyl-containing Ser residue (Fig. 1e). In addition, blue light-dependent phosphorylation of AHA1/AHA2 at Thr-881, AHA5 at Thr-881, and AHA11 at Thr-889 corresponding to AHA1 at Thr-881 was detected (Supplementary Fig. 1). Furthermore, blue light-dependent phosphorylation of the penultimate Thr residue in AHA2 at Thr-947, AHA4/AHA11 at Thr-959/955, and AHA5/8 at Thr-948/947 was also detected[21].

### Phosphorylation of both Thr-881 and Thr-948 is essential for stomatal opening and H⁺-ATPase activation

To address the physiological significance of phosphorylation of AHA1 at Thr-881 and Thr-948 in stomatal responses, transgenic plants expressing the non-phosphorylatable form of AHA1, in which either

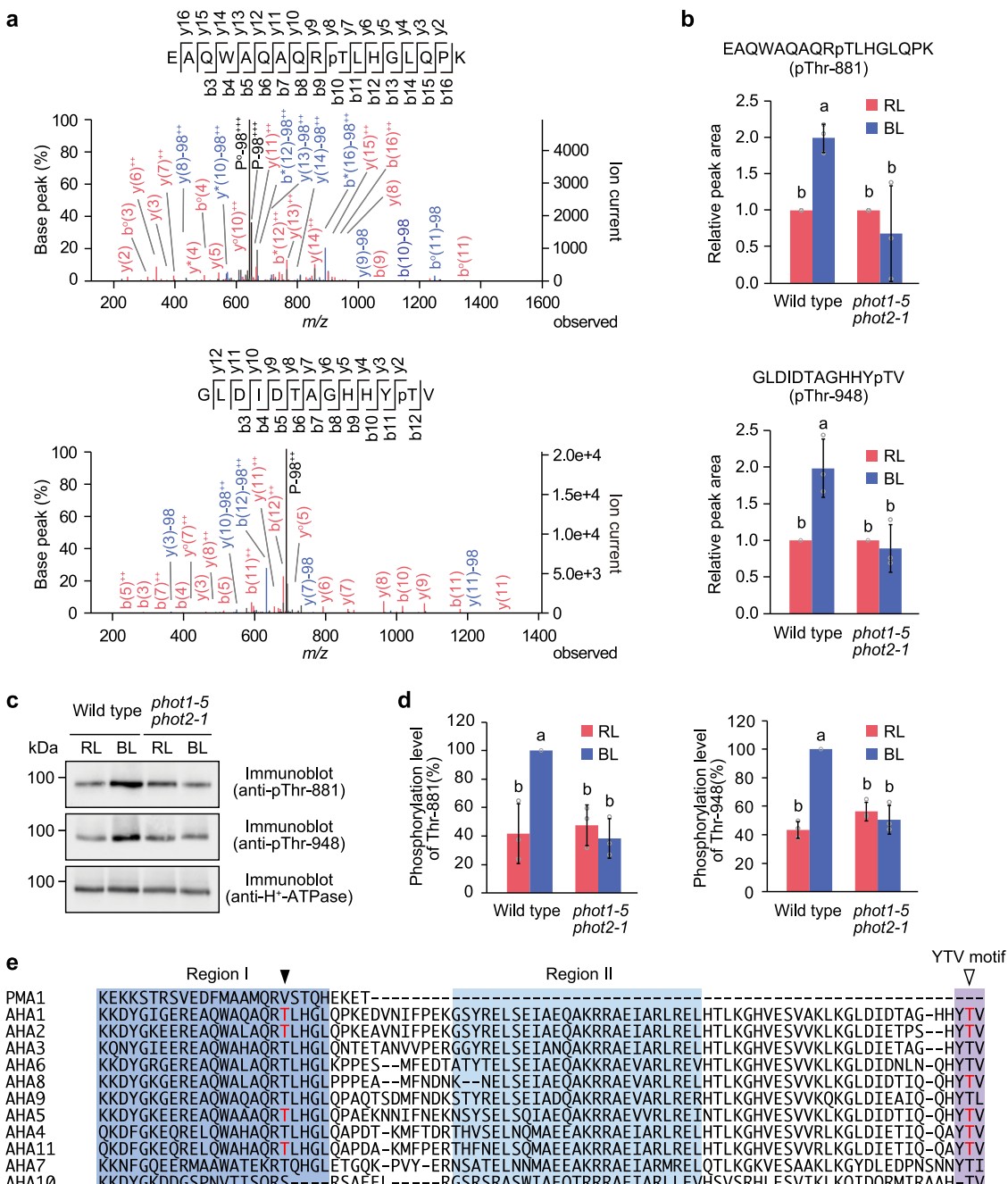

**Fig. 1 | Blue light-dependent phosphorylation of AHA1 at Thr-881 and Thr-948.**
**a** MS/MS spectra of the phosphopeptide of AHA1 containing phospho-Thr-948 and phospho-Thr-881. The labels "−98", "*", and "°" represent neutral loss of $H_3PO_4$, $NH_3$, and $H_2O$ from b, y, and precursor (P) ions, respectively. **b** Quantification of phosphopeptide levels calculated from integrated peak values. The data represent means ± SD ($n = 3$ biologically independent samples). Different letters indicate significant differences (One-way ANOVA with Tukey's test, $P < 0.05$).
**c** Phosphorylation of AHA1 at Thr-881 and Thr-948. Guard cell protoplasts were illuminated with red light (RL: 300 µmol $m^{-2}$ $s^{-1}$) for 30 min, followed by a pulse of blue light (BL: 100 µmol $m^{-2}$ $s^{-1}$, 30 s) superimposed on RL. The phosphorylation and amount of $H^+$-ATPase were detected by immunoblot analysis using anti-pThr881-AHA1, anti-pThr948-AHA1, and anti-$H^+$-ATPase antibodies. **d** Relative

phosphorylation levels of Thr-881 and Thr-948 were quantified using the ImageJ software. Each value is expressed as a percentage of the phosphorylation level of the wild type under BL. Data represent means ± SD ($n = 3$ biologically independent samples). Different letters indicate significant differences (One-way ANOVA with Tukey's test, $P < 0.01$). **e** Alignment of amino acid sequences of the C-terminal regulatory domain of plasma membrane $H^+$-ATPase in *Saccharomyces cerevisiae* (PMA1) and *Arabidopsis thaliana* (AHA1-11). Region I, Region II, and YTV motifs are highlighted in blue, light blue, and purple, respectively. The closed and open arrowheads indicate the positions of Thr-881 and Thr-948 in AHA1, respectively. The blue light-dependent phosphorylation sites identified in this study are indicated in red.

Thr-881 or Thr-948 was replaced by Ala (designated T881A or T948A) in the *aha1-9* null mutant background, were constructed. Consistent with previous reports[38], our immunoblot analysis showed that the total protein levels of $H^+$-ATPase in the guard cells were reduced to 30% in the *aha1-9* mutant compared with those in the wild-type

(Supplementary Fig. 2). Transgenic lines with AHA1 protein levels in guard cells comparable to those in the wild-type plants were selected (Supplementary Fig. 2), and the results confirmed that the stomatal length, density, and index in these transgenic lines were not different from those in the wild-type and *aha1-9* mutant (Supplementary Fig. 3).

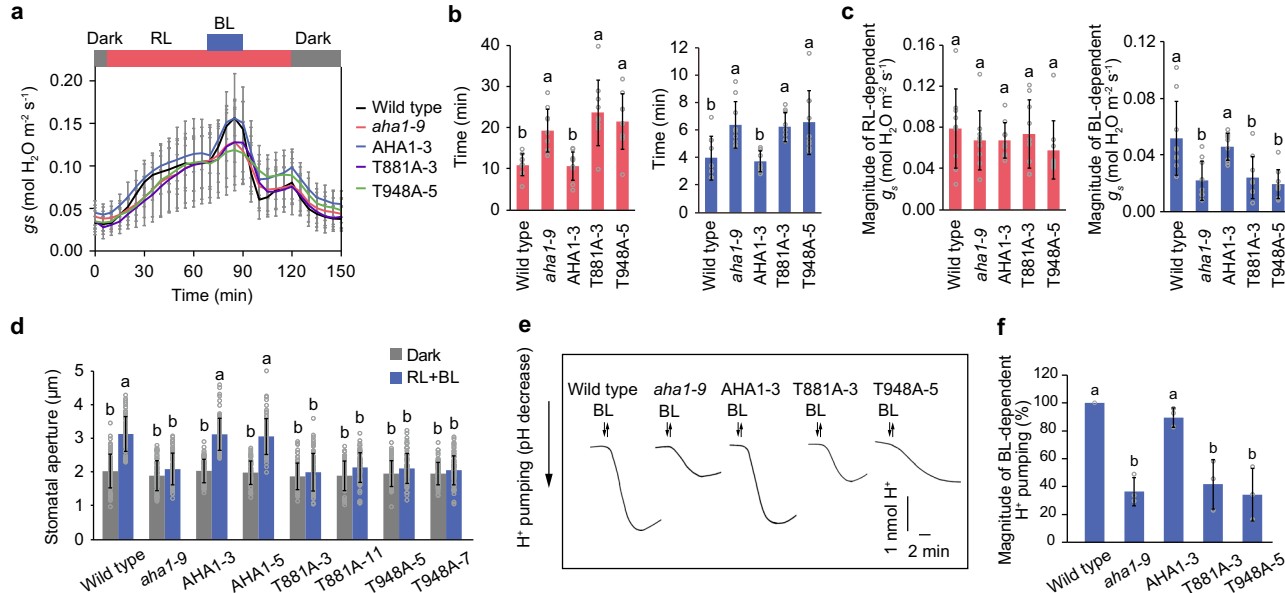

**Fig. 2 | Impairments of blue light-dependent stomatal opening and H⁺ pumping in phosphodefective mutants of Thr-881 and Thr-948 of AHA1. a** Light-dependent changes in stomatal conductance in the intact leaves. Leaves of dark-adapted plants were illuminated with red light (RL: 600 μmol m⁻² s⁻¹) for 1 h, after which blue light (BL: 10 μmol m⁻² s⁻¹) was superimposed on RL for 20 min. The data represent means ± SD ($n = 9$ biologically independent plants). **b** Time taken to reach 30% of the maximum values of stomatal conductance in response to RL and BL. The data represent means ± SD ($n = 9$ biologically independent plants). Different letters indicate significant differences (One-way ANOVA with Tukey's test, $P < 0.05$). **c** Quantification of stomatal conductance changes in response to RL and BL. The data represent means ± SD ($n = 9$ biologically independent plants).

Different letters indicate significant differences (One-way ANOVA with Tukey's test, $P < 0.05$). **d** Light-dependent stomatal opening. Epidermal strips were incubated in the dark or under RL (50 μmol m⁻² s⁻¹) with BL (10 μmol m⁻² s⁻¹) for 2 h. The data represent means ± SD ($n = 75$ stomata from three independent experiments). Different letters indicate significant differences (One-way ANOVA with Tukey's test, $P < 0.01$). **e** Blue light-dependent H⁺ pumping. Guard cell protoplasts were illuminated with RL (300 μmol m⁻² s⁻¹) for 2 h, after which a BL pulse (100 μmol m⁻² s⁻¹, 30 s) was superimposed. **f** Quantification of the magnitude of H⁺ pumping. The data represent means ± SD ($n = 3$ biologically independent experiments). Different letters indicate significant differences (One-way ANOVA with Tukey's test, $P < 0.01$).

Light-dependent stomatal opening in the intact leaves was examined using gas exchange measurements (Fig. 2a–c). A high fluence rate of red light resulted in stomatal opening in the wild-type, and superimposition of a low fluence rate of blue light on the background red light elicited further stomatal opening. In contrast, the rate of stomatal opening induced by red light decreased in the *aha1-9* mutant compared to that in the wild type (Fig. 2b). The *aha1-9* mutant also showed a decrease in both the rate and magnitude of stomatal opening in response to blue light (Fig. 2b, c). Transformation of *aha1-9* with the wild-type *AHA1* gene rescued the stomatal opening induced by red and blue light. However, the introduction of either *T881A* or *T948A* into *aha1-9* failed to restore the stomatal opening in response to red and blue light. The stomatal aperture in the isolated epidermis of phosphodefective lines was further evaluated (Fig. 2d). Irradiation with red and blue light increased the stomatal aperture in the wild-type and AHA1 lines, but not in the *aha1-9* mutant. The expression of T881A and T948A did not recover the stomatal opening defects of *aha1-9*.

To determine the impact of phosphodefective mutations on H⁺-ATPase activation, blue light-dependent H⁺ pumping in the guard cell protoplasts was measured (Fig. 2e, f). In the wild type, blue light caused a decrease in the pH of the outer medium of guard cell protoplasts. Similar to stomatal opening, H⁺ pumping was attenuated by blue light in the *aha1-9* mutant, as well as in the T881A and T948A lines compared with that in the wild-type and AHA1-expressing lines. Together, these findings suggested that the phosphorylation of both Thr-881 and Thr-948 by blue light is essential for stomatal opening and activation of H⁺-ATPase.

### Phosphorylation of Thr-881 by blue light depends on the phosphorylation state of Thr-948

Studies have shown that phot1 and phot2 redundantly induce the phosphorylation of the penultimate Thr residue of H⁺-ATPase

through signalling pathways mediated by BLUS1 and PP1[16,18,40]. Therefore, we examined whether the phosphorylation of Thr-881 by blue light is mediated by a common signalling pathway with Thr-948 phosphorylation. Blue light-dependent phosphorylation of Thr-881 and Thr-948 was not detected in the *phot1 phot2* double mutant (Fig. 1c, d), but was observed in the *phot1* and *phot2* single mutants, although their phosphorylation was more significantly reduced in the *phot1* mutant than in the *phot2* mutant (Fig. 3a). Phosphorylation of Thr-881 and Thr-948 was absent in the guard cells of the *blus1* mutant (Fig. 3b). Furthermore, tautomycin, an inhibitor of PP1, suppressed the phosphorylation of both Thr residues in response to blue light (Fig. 3c).

To study the phosphorylation mechanism of Thr-881 and Thr-948 by blue light in more detail, we determined whether their phosphorylation by blue light depends on their phosphorylation state. Thr-948 phosphorylation was apparent in the T881A line as well as in the wild-type and AHA1 line (Fig. 3d, e). The binding of 14-3-3 proteins to the phosphorylated penultimate Thr-948 analysed by protein blotting was not affected by the T881A mutation in response to blue light (Supplementary Fig. 4). In contrast, Thr-881 phosphorylation in the T948A line was much lower than that in the wild-type and AHA1 line (Fig. 3f, g). These results suggested that Thr-948 phosphorylation was a prerequisite for blue light-dependent Thr-881 phosphorylation. Consistent with these results, we observed that the phosphorylation of Thr-881 was slower than that of Thr-948 in response to blue light. We determined the phosphorylation of both Thr residues in the guard cells after blue light pulse (Fig. 3h, i). The phosphorylation of Thr-948 became evident 30 s after the start of the pulse; subsequently, it gradually increased and reached a maximum at 90 s. In contrast, the phosphorylation of Thr-881 was detected 60 s after blue light and reached a maximum at 90 s.

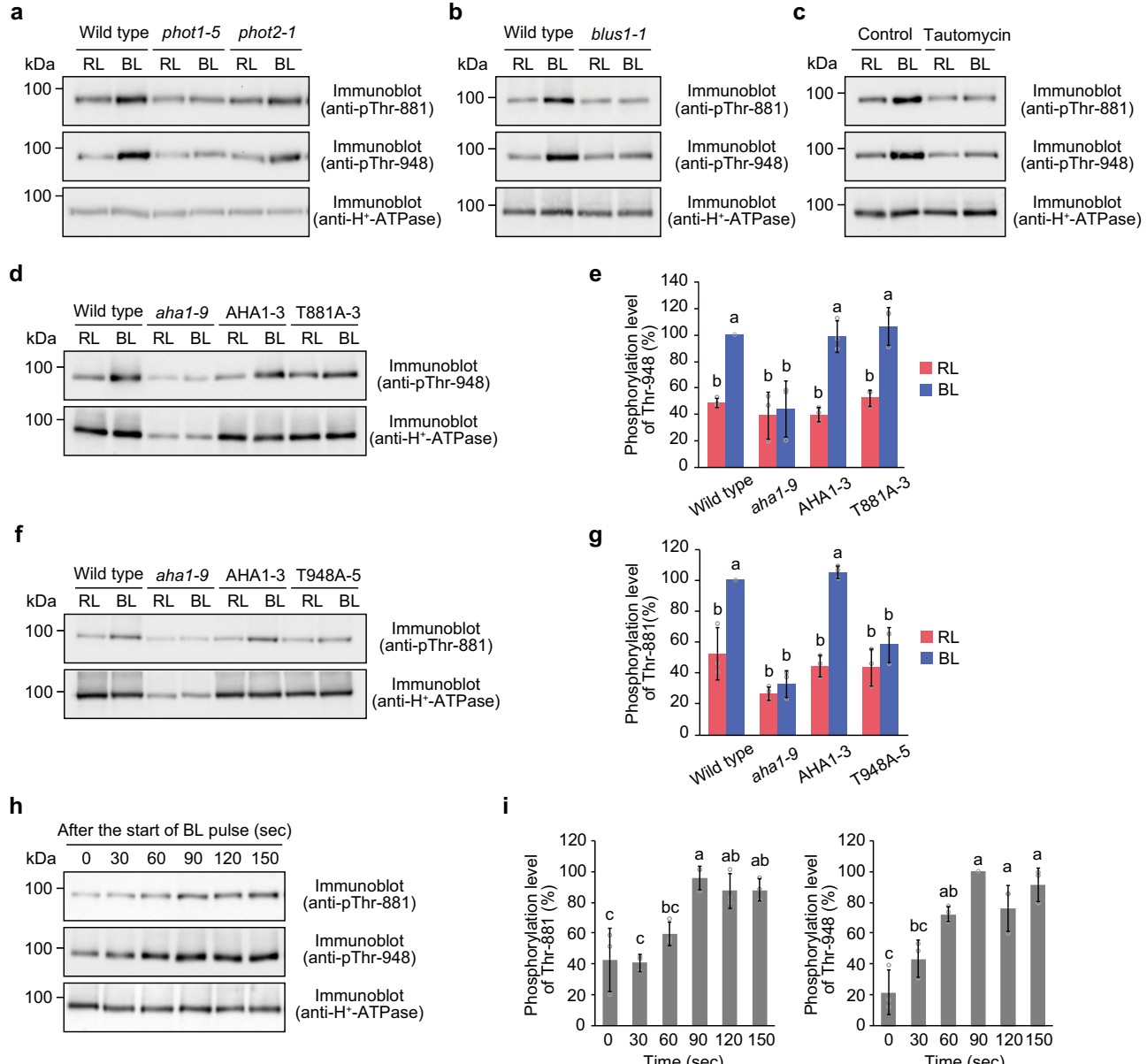

**Fig. 3 | Biochemical analysis of Thr-881 and Thr-948 phosphorylation in response to blue light. a, b** Phosphorylation of Thr-881 and Thr-948 in *phot1-5*, *phot2-1* (**a**), and *blus1-1* mutants (**b**). Guard cell protoplasts were illuminated with red light (RL: 300 μmol m⁻² s⁻¹) for 30 min, after which a pulse of blue light (BL: 100 μmol m⁻² s⁻¹, 30 s) was superimposed on the RL. The reaction was terminated 3 min after the BL pulse. **c** Effect of the PP1 inhibitor tautomycin on Thr-881 phosphorylation. Guard cell protoplasts were pre-treated with tautomycin (5 μM) under RL for 1 h. For (**a**), (**b**), and (**c**), the experiments conducted three times on different occasions gave similar results. **d, e** Phosphorylation of Thr-948 in the T881A mutants. **f, g** Phosphorylation of Thr-881 in the T948A mutants. **h, i** Time

course of Thr-881 and Thr-948 phosphorylations following a BL pulse. The reaction was terminated at the indicated time points. For (**e**), (**g**), and (**i**), the relative phosphorylation levels of Thr-881 and Thr-948 were quantified using the ImageJ software. The data represent means ± SD (*n* = 3 biologically independent experiments). Different letters indicate significant differences (One-way ANOVA with Tukey's test, *P* < 0.01). For (**e**) and (**g**), each value is expressed as a percentage of the phosphorylation level of the wild type under BL. For (**i**), each value is expressed as a percentage of the maximum phosphorylation observed for each phospho-site in response to BL.

## Blue light activation of H⁺-ATPase requires phosphorylation of both Thr-881 and Thr-948

To explore the impact of Thr-881 and Thr-948 phosphorylation on stomatal opening and activation of H⁺-ATPase, we expressed phosphomimetic forms of AHA1 with the substitution of Thr-881 or Thr-948 with the acidic amino acids Asp (D) or Glu (E) in the *aha1-9* mutant. We verified the expression of AHA1 variants and stomatal morphology in the transgenic lines (Supplementary Figs. 5, 6). However, this attempt in Thr-948 was unsuccessful because the T948D and T948E lines failed to complement the stomatal opening in the *aha1-9* mutant (Supplementary Fig. 7). Furthermore, our protein blot analysis revealed that

blue light-dependent binding of 14-3-3 proteins to H⁺-ATPase was abolished in the T948E variant (Supplementary Fig. 8), implying that the T948D and T948E mutations could not adequately mimic the phosphorylation state of Thr-948. In contrast, the T881D line displayed a considerably larger stomatal aperture than that of the wild-type in the dark (Fig. 4a). In accordance with these results, the T881D line exhibited higher ATP hydrolysis activity than that of the wild-type, *aha1-9*, AHA1, and T881A expression lines (Fig. 4b). These findings suggested that the T881D mutation promotes H⁺-ATPase activity.

Furthermore, we found that the T881D line, similar to the wild-type, showed blue light-dependent phosphorylation of Thr-948

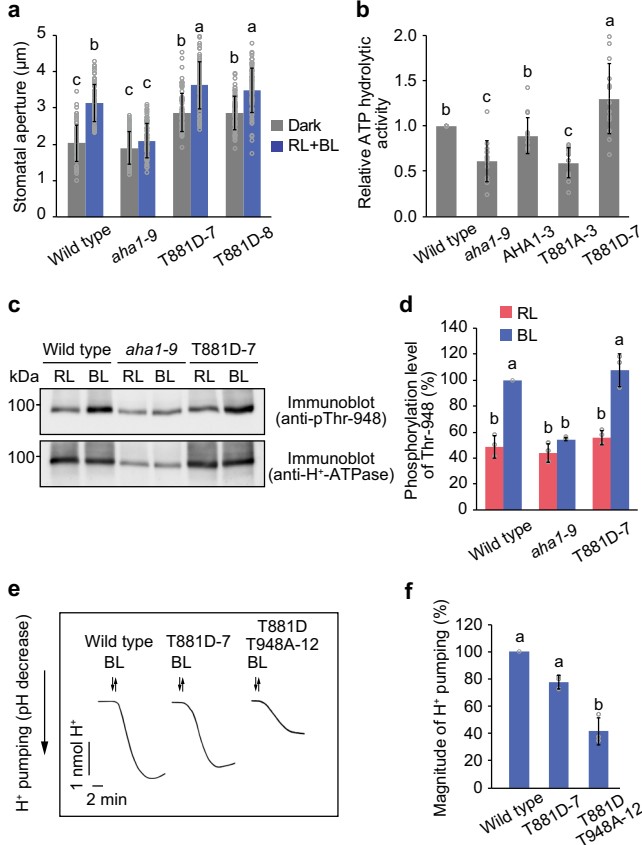

**Fig. 4 | Phosphomimetic mutation of Thr-881 enhances H⁺-ATPase activity.**
**a** Light-dependent stomatal opening. Epidermal strips were incubated in the dark or under red light (RL: 50 μmol m⁻² s⁻¹) with blue light (BL: 10 μmol m⁻² s⁻¹) for 2 h. The data represent means ± SD ($n = 75$ stomata from three independent experiments). Different letters indicate significant differences (One-way ANOVA with Tukey's test, $P < 0.01$). **b** ATP hydrolytic activity in the microsomal membrane fraction obtained from leaves. The data represent means ± SD ($n = 14$ biologically independent experiments). Different letters indicate significant differences (One-way ANOVA with Tukey's test, $P < 0.05$). **c** Phosphorylation of Thr-948 in the T881D line. Guard cell protoplasts were illuminated with RL (300 μmol m⁻² s⁻¹) for 30 min, after which a pulse of BL (100 μmol m⁻² s⁻¹, 30 s) was superimposed. **d** Relative phosphorylation levels of Thr-948 were quantified using the ImageJ software. Each value is expressed as a percentage of the phosphorylation level of the wild type under BL. The data represent means ± SD ($n = 3$ biologically independent experiments). Different letters indicate significant differences (One-way ANOVA with Tukey's test, $P < 0.01$). **e** BL-dependent H⁺ pumping. Guard cell protoplasts were illuminated with RL (300 μmol m⁻² s⁻¹) for 2 h, after which a BL pulse (100 μmol m⁻² s⁻¹, 30 s) was superimposed. **f** Quantification of the magnitude of H⁺ pumping. The data represent means ± SD ($n = 3$ biologically independent experiments). Different letters indicate significant differences (One-way ANOVA with Tukey's test, $P < 0.01$).

(Fig. 4c, d). Furthermore, this line exhibited H⁺ pumping in response to blue light in the guard cell protoplasts (Fig. 4e, f). We created transgenic lines harbouring double mutations of T881D T948A in AHA1 (Supplementary Figs. 5, 6) and found that H⁺ pumping by blue light was abolished in the T881D T948A expressing line. Together, these results suggested that Thr-948 phosphorylation also stimulates H⁺-ATPase activity and that phosphorylation of both Thr-881 and Thr-948 by blue light may ensure full activation of H⁺-ATPase.

### Guard cell photosynthesis mediates Thr-881 phosphorylation
The stomatal response to red light was delayed in the T881A and T948A lines as well as in the *aha1-9* mutant (Fig. 2a, b). Our guard cell phosphorylation analysis showed that Thr-881 is phosphorylated in response to red light in addition to blue light. Following red light

irradiation, the phosphorylation level of Thr-881 increased and was further enhanced by the superimposition of blue light on red light (Fig. 5a, b). However, unlike Thr-881, Thr-948 was not phosphorylated by red light but was phosphorylated in a blue light-dependent manner.

Red light-induced phosphorylation of Thr-881 was completely inhibited after treatment with the photosynthetic electron transport inhibitor 3-(3,4-dichlorophenyl)−1, 1-dimethylurea (DCMU), suggesting that this phosphorylation relies on guard cell photosynthesis (Fig. 5c, d). In contrast, blue light-dependent phosphorylation of Thr-881 was not affected by the DCMU treatment (Fig. 5c, d, and Supplementary Fig. 9). Furthermore, Thr-881 phosphorylation by red light was evident in the *phot1 phot2* and *blus1* mutants (Fig. 5e–h).

We further evaluated whether Thr-948 phosphorylation is required for Thr-881 phosphorylation by red light. Red light-induced Thr-881 phosphorylation was found in the T948A line as well as in the wild-type (Fig. 5i, j), suggesting that Thr-881 phosphorylation by red light does not require Thr-948 phosphorylation. Collectively, these findings suggested that red light-induced Thr-881 phosphorylation relies on guard cell photosynthesis and is distinct from phototropin-mediated blue light signalling.

## Discussion
In this study, we demonstrated that blue light induces the phosphorylation of two Thr residues of the plasma membrane H⁺-ATPase in guard cells, which is crucial for stomatal opening. The phosphoproteome analysis using guard cell protoplasts revealed that Thr-881 and Thr-948, located in the C-terminal autoinhibitory domain of AHA1, were phosphorylated in response to blue light (Fig. 1a, b). The immunoblot analysis using antibodies specific for individual Thr residues showed that these residues were phosphorylated in a blue light- and phototropin-dependent manner (Fig. 1c, d). Substitution of Thr-881 and Thr-948 in AHA1 with Ala abrogated the stomatal opening induced by blue light (Fig. 2a–d). Furthermore, both the T881A and T948A lines showed impairment of blue light-dependent H⁺ pumping from guard cell protoplasts (Fig. 2e, f). These findings are consistent with the growth defects of yeast transformants bearing mutations in which the conserved Thr in region I and penultimate Thr were replaced by Ala[37,41,42]. From these results, we concluded that phosphorylation of both Thr-881 and Thr-948 of AHA1 is essential for blue light-dependent activation of H⁺-ATPase in guard cells.

Numerous previous studies have reported Thr-948 phosphorylation in guard cells in response to blue light[33,43]. In this study, we discovered that Thr-881 is also phosphorylated in guard cells in both blue light- and photosynthesis-dependent manners. A recent study suggested that Thr-881 is phosphorylated in plants in response to low K⁺ treatments[36]. Thus, the phosphorylation level of Thr-881 appears to be regulated as a terminal point in the signalling pathways of various external cues.

Furthermore, we examined the relationship between Thr-881 and Thr-948 phosphorylation on AHA1 in response to blue light. The time-course analysis revealed that compared to that of Thr-948, the start of phosphorylation of Thr-881 was delayed by blue light irradiation in wild-type guard cells (Fig. 3h, i). Furthermore, site-directed mutagenesis showed that blue light-dependent phosphorylation of Thr-948 and binding of 14-3-3 proteins were evident in lines harbouring the T881A mutation (Fig. 3d, e and Supplementary Fig. 4). In accordance with these findings, the phosphorylation of penultimate Thr-955 of *Nicotiana tabacum* plasma membrane H⁺-ATPase 2 (PMA2) was not significantly affected in tobacco BY2 cells expressing PMA2 variants in which Thr-889 corresponding to Thr-881 of AHA1 was replaced by Ala[41]. Similarly, phosphorylation of penultimate Thr-947 was observed when AHA2 with a T881A mutation was expressed in yeast cells[37]. In contrast, Thr-881 phosphorylation under blue light was significantly decreased in the AHA1 T948A line compared to that in the wild-type and AHA1 line (Fig. 3f,

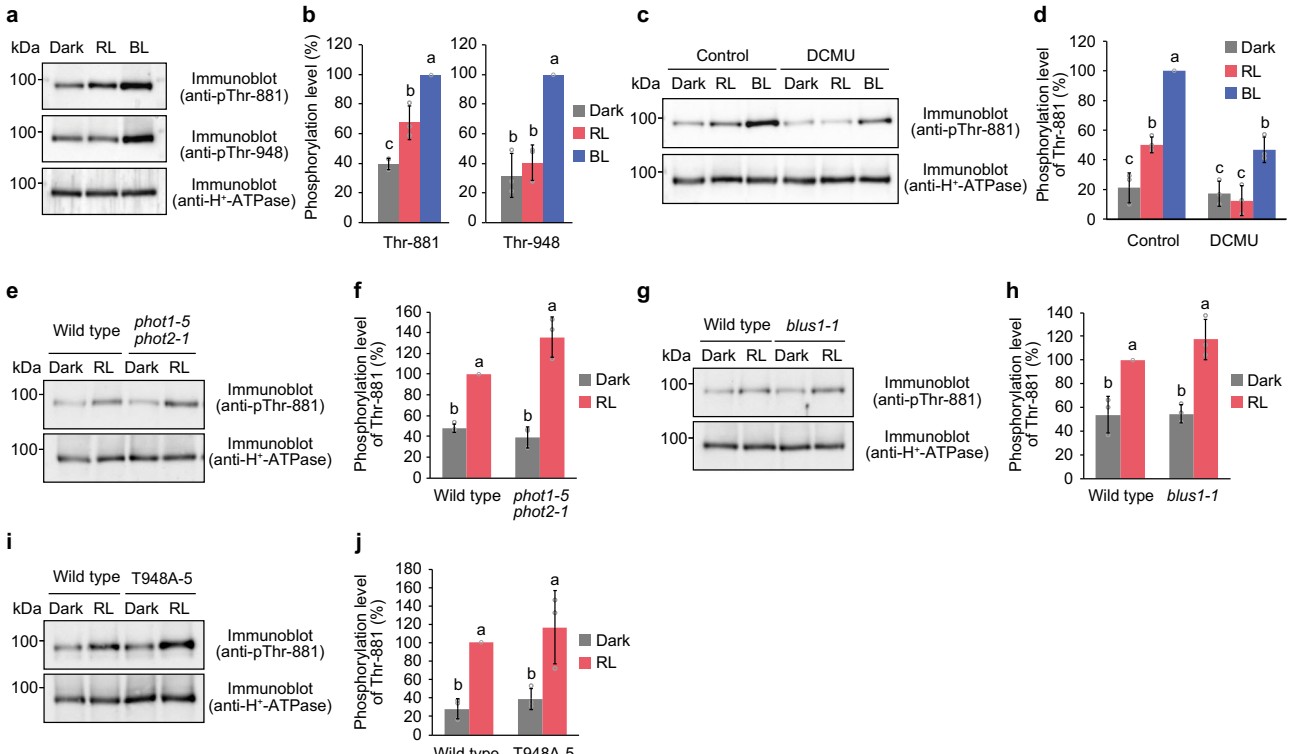

**Fig. 5 | Guard cell photosynthesis induces Thr-881 phosphorylation. a, b** Red light-dependent phosphorylation of Thr-881. Guard cell protoplasts were incubated in the dark for 30 min or illuminated with red light (RL: 300 µmol m⁻² s⁻¹) for 30 min, after which a pulse of blue light (BL: 100 µmol m⁻² s⁻¹, 30 s) was superimposed on RL. **c, d** Effect of the photosynthesis inhibitor 3-(3,4-dichlorophenyl)-1, 1-dimethylurea (DCMU) on Thr-881 phosphorylation. Guard cell protoplasts were pre-treated with DCMU (10 µM) in the dark for 30 min. **e–j** Red light-dependent phosphorylation of Thr-881 in the *phot1-5 phot2-1* (**e, f**), *blus1-1* (**g, h**), and the T948A line (**i, j**). For (**b**), (**d**), (**f**), (**h**), and (**j**), the relative phosphorylation levels of Thr-881 and Thr-948 were quantified using the ImageJ software. The data represent means ± SD (*n* = 3 biologically independent experiments). Different letters indicate significant differences, One-way ANOVA with Tukey's test, *P* < 0.01 for (**b**), (**f**), and (**h**) and *P* < 0.05 for (**d**) and (**j**).

g). Taken together, our findings suggested that Thr-881 phosphorylation by blue light depends on the phosphorylation status of Thr-948.

It remains unclear whether Thr-881 phosphorylation is caused by a secondary response associated with Thr-948 phosphorylation or is directly regulated by blue light signals. Our findings indicated that Thr-881 phosphorylation depends on Thr-948 phosphorylation (Fig. 3f, g), suggesting the possibility that blue light signals are transmitted to Thr-948 to facilitate its phosphorylation, subsequently leading to Thr-881 phosphorylation because of conformational changes coupled with Thr-948 phosphorylation without blue light signals. However, besides the requirement of Thr-948 phosphorylation as a prerequisite for Thr-881 phosphorylation, the possibility that Thr-881 may also undergo phosphorylation by protein kinases activated through blue light signals cannot be ruled out. Further research is needed to fully elucidate the mechanisms underlying blue light regulation of Thr-881 phosphorylation.

Our results showed that the T881D line displayed enhanced stomatal opening in epidermal tissues in the dark (Fig. 4a). Furthermore, the microsomal fraction isolated from the T881D line showed increased ATP hydrolytic activity (Fig. 4b). Such increased ATP hydrolysis activity was also observed in yeast cells expressing the phosphomimetic PMA2 T899D variant[41]. Thus, the phosphorylation of Thr-881 in AHA1 appears to promote H⁺-ATPase activity. In addition, Thr-948 phosphorylation was prominent when the T881D cell line was exposed to blue light (Fig. 4c, d). Furthermore, the T881D line showed blue light-dependent H⁺ pumping activity, whereas the blue light response in the T881D line was abrogated by introducing an additional T948A mutation (Fig. 4e, f). Consistent with our findings, the

expression of AHA2 T881D and PMA2 T889D increased yeast growth in low-pH media, but such growth enhancement was lost in lines expressing AHA2 T881D T947A and PMA2 T889D T955A[37,41]. From these results, we concluded that the simultaneous phosphorylation of Thr-881 and Thr-948 by blue light induces full activation of H⁺-ATPase.

The functional impact of Thr-881 and Thr-948 phosphorylation on H⁺-ATPase activity remains unknown. Previous Ala scanning mutagenesis experiments and expression of various C-terminal truncated constructs indicated that regions I and II in the C-terminal region of H⁺-ATPase play an important role in the autoinhibition of catalytic activity[44]. Treatment of the plasma membrane with trypsin removes the C-terminal segment from H⁺-ATPase, resulting in the activation of H⁺ pumping in the plasma membrane vesicles[30]. The synthetic peptide containing region I inhibits H⁺ pumping by trypsin-activated H⁺-ATPase[30], implying an interaction between region I peptide and the rest of the C-terminally truncated H⁺-ATPase. Indeed, a yeast interaction assay confirmed the binding of the C-terminal region to the central loop of plasma membrane H⁺-ATPase[31]. In vivo cross-linking experiments revealed inter- and intramolecular interactions of amino acids between the C-terminal and the catalytic domain of H⁺-ATPase[32,45]. Furthermore, recent cryo-electron microscopy studies of yeast and *Neurospora crassa* PMA1 revealed that the C-terminal peptide, including region I, forms an α helix structure that mediates the interaction between the P-domain and the neighbouring P-domain via salt bridges in the autoinhibited state[46,47]. In contrast, the C-terminal helix becomes disordered in the active state, enabling the movement of the P-domain, which is essential for ATP catalysis[47]. The C-terminal regulatory helix of PMA1 contains three phosphorylation sites, namely Ser-899, Ser-911, and Thr-912. Although no structural information of

the phosphorylated C-terminal domain is available, the structural analysis described above suggested that these phosphorylations may alleviate autoinhibition by disrupting salt bridges and suppressing the helix from binding to the P-domain[47]. In the present study, the sequence alignment showed that Thr-881 in AHA1 is closely related to Ser-911 and Thr-912 in PMA1 (Fig. 1e). Therefore, phosphorylation of Thr-881 by blue light is likely to modulate the helical structure of region I and its inhibitory interactions. On the other hand, phosphorylation of the penultimate Thr residue creates a binding site, Tyr-pThr-Val, for 14-3-3 proteins[34]. Such binding of 14-3-3 proteins to the C-terminal sequence appears to cause the conversion of H⁺-ATPase from a dimer to a hexamer, enhancing its activity and stabilising its activated state[35,48]. Thus, phosphorylation of Thr-881 and Thr-948 may promote H⁺-ATPase activation via different modes of regulation, which is in accordance with our finding that simultaneous phosphorylation of both Thr residues is required for full activation of H⁺-ATPase.

Our results indicated that Thr-881 in AHA1 is phosphorylated not only by blue light but also by red light in guard cell protoplasts. Red light-induced phosphorylation of Thr-881 was inhibited by DCMU, suggesting a requirement for guard cell photosynthesis (Fig. 5c, d). In contrast, no significant increase in Thr-948 phosphorylation was observed in guard cell protoplasts under red light (Fig. 5a, b). It has been demonstrated that red light stimulates H⁺ pump activation at the plasma membrane in *Vicia faba* guard cells and that this response is inhibited by DCMU[49]. Therefore, it is plausible that red light stimulates plasma membrane H⁺-ATPase by phosphorylating Thr-881 through guard cell photosynthesis.

We should note here that Thr-948 phosphorylation in the H⁺-ATPase by red light differs between isolated guard cell protoplasts and guard cells in intact leaves. In guard cell protoplasts, Thr-948 phosphorylation does not occur by red light. In leaves, Thr-948 phosphorylation in guard cells was observed when the leaves were illuminated with red light[50]. This red light-induced Thr-948 phosphorylation in leaves was suppressed by DCMU[50], suggesting possible involvement of mesophyll cell photosynthesis in this process. The discrepancy in Thr-948 phosphorylation by red light between isolated protoplasts and intact leaves is probably related to the presence of diffusible signals derived from mesophyll cell photosynthesis in leaves[51]. Among the potential mesophyll signals that induce Thr-948 phosphorylation in guard cells, sucrose produced by mesophyll cell photosynthesis appears to be the most likely candidate. This idea was supported by a study showing that the exogenous application of sucrose to guard cell protoplasts induced Thr-948 phosphorylation[52]. In contrast, a recent investigation revealed that subjecting leaves to elevated $CO_2$ concentration resulted in Thr-948 dephosphorylation in guard cells[53]. This implies that the reduction in intercellular $CO_2$ concentration within leaves by mesophyll cell photosynthesis could induce Thr-948 phosphorylation in guard cells.

Importantly, red light-induced stomatal opening has been observed to be delayed in the *aha1-9* mutant[38,50], indicating that plasma membrane H⁺-ATPase also plays a role in red light-induced stomatal opening, in addition to its role in blue light-dependent stomatal opening. However, direct evidence regarding the necessity of Thr-881 and Thr-948 phosphorylation in red light-induced stomatal opening has not yet been obtained. Our data revealed that both the T881A and T948A lines showed delayed stomatal opening by red light, similar to the *aha1-9* mutant (Fig. 2a, b). These findings, along with other existing evidence[50], suggest that mechanisms regulating Thr-881 and Thr-948 phosphorylation in leaf guard cells by red light are distinct, with Thr-881 phosphorylation being mediated via guard cell photosynthesis and Thr-948 phosphorylation being mediated via mesophyll cell photosynthesis. Moreover, both of these phosphorylation events may contribute to the process of red light-induced stomatal opening.

Stomata open synergistically when exposed to blue light and high-intensity red light, which drives photosynthesis[6,8]. Guard cells sense the information of both blue light and photosynthetic activities and integrate it for proper control of stomatal opening. Such mechanisms are crucial for plants to ensure accurate gas exchange for photosynthetic $CO_2$ fixation while preventing water loss under changing light environments. Our previous findings indicated that plasma membrane hyperpolarisation, achieved by the activation of H⁺-ATPase by blue light and inactivation of S-type anion channels by the decrease in intercellular $CO_2$ concentration via photosynthetic $CO_2$ fixation, acts as a key mechanism for the fine control of stomatal opening by blue and red light[12]. In addition to these mechanisms, coordinated control of the phosphorylation levels of guard cell plasma membrane H⁺-ATPase by guard cell and mesophyll cell photosynthesis and blue light signals may represent a key determinant of the synergistic action of blue and red light on stomatal opening. Notably, blue light also causes dephosphorylation of guard cell H⁺-ATPase, which probably functions as a negative feedback regulator of stomatal opening[12].

In this study, different phosphorylation patterns of Thr-881 and Thr-948 in guard cell protoplasts in response to blue and red light were observed. Specifically, blue light elicited phosphorylation of both Thr-881 and Thr-948, whereas red light only induced phosphorylation of Thr-881 in guard cell protoplasts (Figs. 1a, b and 5a, b). Mutants of *phot1 phot2* and *blus1* lost the phosphorylation of Thr-881 and Thr-948 in response to blue light, whereas they exhibited Thr-881 phosphorylation in response to red light (Figs. 1a–d, 3b and 5e–h). Furthermore, the T948A line exhibited impaired blue light-dependent phosphorylation of Thr-881, but it displayed red light-induced phosphorylation of Thr-881 (Figs. 3f, g and 5i, j). This suggests that Thr-948 phosphorylation is a prerequisite for blue light-induced Thr-881 phosphorylation but is not essential for red light-induced Thr-881 phosphorylation. Furthermore, blue light-induced Thr-881 and Thr-948 phosphorylation remained unaffected by DCMU, whereas red light-induced Thr-881 phosphorylation was inhibited by DCMU (Fig. 5c, d and Supplementary Fig. 9)[54]. Based on these findings, we concluded that Thr-948 phosphorylation in guard cell protoplasts is regulated by phototropin-mediated blue light signalling, whereas Thr-881 phosphorylation in response to blue and red light is regulated by separate signalling pathways originating from phototropin and guard cell photosynthesis, respectively (Fig. 6).

The protein kinases responsible for the phosphorylation of Thr-881 and Thr-948 in response to blue and red light remain unclear. Receptor-like transmembrane kinases (TMKs) were recently shown to be involved in auxin-induced phosphorylation of the penultimate Thr residue of H⁺-ATPase[55,56]. It would be pertinent to investigate whether TMKs are responsible for Thr-881 and Thr-948 phosphorylation in guard cells in response to blue and red light. In contrast, D-clade type 2 C protein phosphatases (PP2C-D) have been implicated in the dephosphorylation of the penultimate Thr[57,58]. Furthermore, the leucine-rich repeat receptor-like kinases BRI1-ASSOCIATED RECEPTOR KINASE 1 (BAK1) and PSY1-RECEPTOR (PSY1R) directly bind to and phosphorylate AHA2 at Thr-881[36,37,59]. The Thr-881 of AHA2 is phosphorylated by PLANT PEPTIDE CONTAINING SULFATED TYROSINE 1 (PSY1) in *Arabidopsis* seedlings, but not in the *psy1r* mutant[37]. In contrast, a recent study demonstrated that the phosphorylation level of AHA2 at Thr-881 is not increased by the addition of PSY1 and PSY5 to *Arabidopsis* roots[60]. Phosphorylation of Thr -881 by PSY peptides is controversial. To gain a deeper understanding of the molecular functions of Thr-881 and Thr-948 phosphorylation in light-induced stomatal opening, it is necessary to identify the protein kinases and phosphatases that regulate the phosphorylation at these critical sites in guard cells.

In addition to Thr-881 and Thr-948, several phosphorylation sites in the C-terminal autoinhibitory domain that affect H⁺-ATPase activity have been identified[61,62]. PROTEIN KINASE SOS2-LIKE5 (PKS5)

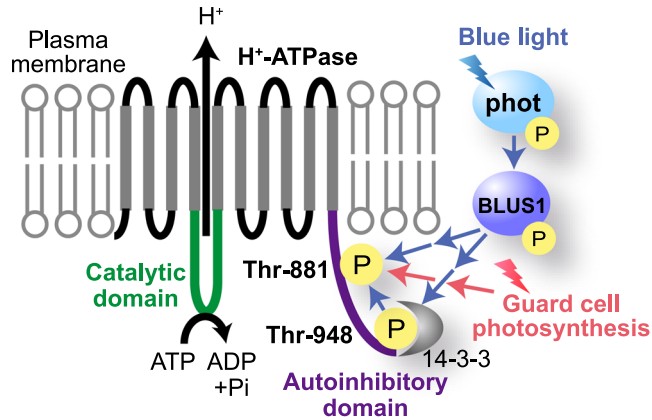

**Fig. 6 | Schematic model for blue light and photosynthesis-mediated activation of H⁺-ATPase in the guard cells.** Blue light-activated phototropins phosphorylate BLUS1 protein kinase, which mediates the phosphorylation of Thr-881 and Thr-948. Thr-948 phosphorylation is a prerequisite for Thr-881 phosphorylation in response to blue light. Simultaneous phosphorylation of Thr-881 and Thr-948 promotes H⁺-ATPase activity and stomatal opening. Guard cell photosynthesis also mediates Thr-881 phosphorylation.

phosphorylates AHA2 at Ser-931 and inhibits the activation of H⁺-ATPase by preventing the binding of 14-3-3 proteins, regardless of the phosphorylation state of the penultimate Thr residue[63]. Similarly, phosphorylation of AHA2 at Ser-899 following the application of the peptide hormone RAPID ALKALIZATION FACTOR (RALF) results in the inactivation of H⁺-ATPase via unknown mechanisms[64]. Furthermore, a recent study showed that BAK1 interacts with AHA2 and phosphorylates Ser-944, consequently stimulating AHA2 activity[59]. It would be interesting to investigate whether these sites are phosphorylated in response to other stimuli in the guard cells.

In conclusion, the present study provides insights into the molecular mechanisms underlying the light regulation of plasma membrane H⁺-ATPase and stomatal opening. Considering that both Thr-881 and Thr-948 are conserved in most land plants[62], the activation of H⁺-ATPase via the phosphorylation of these two Thr residues may be an evolutionarily conserved mechanism beyond light signalling in the guard cells. Elucidation of the regulatory mechanisms of Thr-881 and Thr-948 phosphorylation, using guard cells as a model case, will provide important insights into how plants control proton gradients across the plasma membrane, cytosolic pH, and cell turgor by various environmental and endogenous signals.

## Methods
### Plant materials and growth conditions
In this study, *Arabidopsis thaliana* accession Col-0 and previously described mutants of *phot1-5*[65], *phot2-1*[66], *phot1-5 phot2-1*[40], *blus1-1*[16], and *aha1-9*[38] were used. The plants were grown in a mixture of soil and vermiculite (1:1) for four weeks under white light (50 µmol m⁻² s⁻¹) with a 14/10 h light/dark cycle at 24 °C.

### Phosphoproteome analysis
Phosphoproteome analyses using guard cell protoplasts from the wild-type and *phot1-5 phot2-1* mutant were performed as described previously[16,21]. Briefly, the guard cell protoplasts were incubated under red light (600 µmol m⁻² s⁻¹) for 30 min, after which a pulse of blue light (100 µmol m⁻² s⁻¹, 30 s) was superimposed on the background red light (*n* = 3 biologically independent samples). The reaction was terminated by adding trichloroacetic acid to the protoplast suspension 2.5 min after the start of blue light illumination. The guard cell proteins were dissolved in 0.1 M Tris-HCl (pH 9.0) containing 8 M urea, protease inhibitor cocktail (Sigma), and phosphatase inhibitor cocktails 1 and 2

(Sigma). After a reduction with dithiothreitol and carbamidomethylation with iodoacetamide, the samples were digested with Lys-C and trypsin, and phosphopeptides were enriched by hydroxy acid-modified metal oxide chromatography (HAMMOC) using lactic acid-modified titania[39,67]. The resulting phosphopeptides were eluted and analysed with nanoLC/MS/MS analyses (LTQ-Orbitrap, Thermo Fisher Scientific) in Data-Dependent Acquisition (DDA) mode under the following conditions.

The mass spectrometer was coupled with Ultimate3000 pump (Thermo Fisher Scientific) and an HTC-PAL autosampler (CTC Analytics). Reprosil-Pur 120 C18-AQ beads (3 µm; Dr. Maisch GmbH) packed in a self-pulled needle (150 mm length × 100 µm i.d., 6-µm opening) were used as the nanoLC column. The mobile phases consisted of 0.5% acetic acid (A) and 0.5% acetic acid and 80% acetonitrile (B). A three-step linear gradient of 5% to 10% B in 5 min, 10% to 40% B in 60 min, 40% to 100% B in 5 min, and 100% B for 10 min was applied, operating at a flow rate of 500 nL/min. A spray voltage of 2400 V was applied. The full MS scan was acquired by Orbitrap from *m/z* 300 to 1500 with a resolution of 60,000. The top-10 precursor ions were selected for subsequent MS/MS scans by ion trap in the automated gain control mode, where the automated gain control values of 5.00e + 05 and 1.00e + 04 were set for full MS and MS/MS, respectively. The normalized collision-induced dissociation was set to 35.0.

Peptides and proteins were identified by database search using Mascot version 2.3 (Matrix Science) against the TAIR database (release 10) with trypsin specificity allowing up to two missed cleavages. The precursor mass tolerance and fragment ion mass tolerance were set as 3 ppm and 0.8 Da, respectively. Carbamidomethylation of cysteines was set as the fixed modification. Oxidation of methionines and phosphorylation of serines, threonines, and tyrosines were set as the variable modifications. Peptide hits were accepted if the Mascot score was over the 95% confidence limit based on the "identity" score of each peptide and if at least three successive y- or b-ions with a further two or more y-, b-, and/or precursor-origin neutral loss ions were observed and the minimum peptide length was set to 6. A decoy database search against randomized sequences estimated less than 1% false-positive rate for the identified phosphopeptides with these criteria. Localisation of phosphorylated sites in the identified phosphopeptides was confirmed in-house Perl script to check for the presence of a site-determining ion combination[68]. The XIC peak area of the identified phosphopeptide was integrated using Mass Navigator v1.2 (Mitsui Knowledge Industry).

### Generation of transgenic plants
Transgenic plants expressing various AHA1 variants were generated as previously described with modifications[38]. The genomic sequence containing the promoter, coding region, and 3′ noncoding region of *AHA1* was amplified in two fragments using the following primers: 5′-GGCCAGTGCCAAGCTTCTACTACACATACATGAGTC-3′ and 5′-GTAGCAATCAGAATTCACAAGTTGCTTCTACTGATAG-3′ for the upstream sequence (5106 bp) and 5′-CTTGATATCGAATTCCTCTATTGTAATTGATTTTGTTTAGTGAAATTG-3′ and 5′-TCTAGAACTAGTGGATCCCATCCATATCTTTGGACG-3′ for the downstream sequence (5602 bp). The resulting upstream sequence was subcloned into the *Hind* III/*Eco* RI site of the pRI 101-AN vector (TaKaRa) in which the kanamycin resistance gene was replaced with the hygromycin resistance gene. The downstream sequence was subcloned into the *Eco* RI/*Bam* HI site of the pBluescript SK (+) vector (Stratagene) using the In-Fusion HD cloning kit (Clontech). The latter vector, containing the downstream sequence, was used as a template for point mutations using the QuikChange Site-Directed Mutagenesis Kit (Stratagene). The primers used were 5′-GGGCACAAGCTCAAAGGGCATTGCACGGTCTGCAGCC-3′ and 5′-GGCTGCAGACCGTGCAATGCCCTTTGAGCTTGTGCCC-3′ for T881A, 5′-GGGCACAAGCTCAAAGGGACTTGCACGGTCTGCAGCC-3′ and 5′-GGCTGCAGACCGTGCAAGTCCCTTTGAGCTTGTGCCC-3′

for T881D, 5′-GCAGGACATCACTACGCTGTGTAGTTGGAGTTGCACAA-CAAC-3′ and 5′-GTTGTTGTGCAACTCCAACTACACAGCGTAGTGATGT CCTGC-3′ for T948A, 5′-GCAGGACATCACTACGATGTGTAGTTGGA GTTGCACAACAAC-3′ and 5′-GTTGTTGTGCAACTCCAACTACACATC GTAGTGATGTCCTGC-3′ for T948D, and 5′-GCAGGACATCACT ACGAAGTGTAGTTGGAGTTGCACAACAAC-3′ and 5′-GTTGTTGTGCAA CTCCAACTACACTTCGTAGTGATGTCCTGC-3′ for T948E. After introducing mutations, the downstream sequence was amplified using the primers 5′- AGCAACTTGTGAATTCCTCTATTGTAATTGATTTTG-3′ and 5′-GTAGCAAT- CAGAATTCCATCCATATCTTTGGACGTG-3′ and inserted into the *Eco* RI site of the pRI 101-AN vector containing the upstream sequence. The resulting constructs were transformed into the *aha1-9* mutant using *Agrobacterium tumefaciens* strain GV3101.

### Isolation of guard cell protoplasts and measurement of H⁺ pumping

The guard cell protoplasts were enzymatically isolated from the fully developed leaves of 4-week-old *Arabidopsis* plants as described previously[19,43]. Blue light-dependent H⁺ pumping from the guard cell protoplasts was measured using a glass pH electrode as described previously[43]. The guard cell protoplasts were incubated in 0.125 mM MES-NaOH (pH 6.0), 1 mM $CaCl_2$, 0.4 M mannitol, and 10 mM KCl under red light (300 µmol $m^{-2}$ $s^{-1}$) for 2 h at 24 °C. Thereafter, a pulse of blue light (100 µmol $m^{-2}$ $s^{-1}$, 30 s) was superimposed on the background red light.

### Immunoblot analysis of guard cell proteins

To determine the phosphorylation of plasma membrane H⁺-ATPase, the guard cell protoplasts were incubated in 0.125 mM MES-NaOH (pH 6.0), 1 mM $CaCl_2$, 0.4 M mannitol, and 10 mM KCl under red light (300 µmol $m^{-2}$ $s^{-1}$) for 30 min at 24 °C, after which a pulse of blue light (100 µmol $m^{-2}$ $s^{-1}$, 30 s) was superimposed on the background red light. The reaction was terminated 3 min after the start of blue light illumination by adding trichloroacetic acid to the protoplast suspension. For tautomycin treatments, the guard cell protoplasts were preincubated under red light for 30 min, after which tautomycin was added to the final concentration of 5 µM. The protoplasts were further incubated under red light for 1 h and irradiated with a blue light pulse. For red light-induced phosphorylation of H⁺-ATPase, the guard cell protoplasts were preincubated in the dark for 30 min and then illuminated under red light for 30 min. For the DCMU treatment, the guard cell protoplasts were preincubated in the dark for 10 min and further incubated in the dark for 30 min in the presence of 10 µM DCMU before red light illumination.

Immunoblotting was performed as described previously[19,33] with slight modifications. Antibodies against the plasma membrane H⁺-ATPase were generated using the GST fusion protein of the hydrophilic loop of AHA1 (M320-A608) according to Kinoshita and Shimazaki (1999)[33]. Phospho-specific antibodies against Thr-881 and Thr-948 of AHA1 were generated using the synthetic peptides AQAQRpTLHGLQPKE for Thr-881 and IDTAGHHYpTV for Thr-948. Antibodies against BLUS1 have been described previously[16]. The phosphorylation of Thr-948 was also determined by protein blotting using GST-14-3-3[16,19]. Intensity of the protein bands was quantified using the ImageJ 1.48 software (National Institutes of Health).

### Measurement of stomatal opening

Stomatal conductance ($g_s$) of intact leaves was measured using a gas-exchange system (Li-6400; Li-Cor) under the following conditions: 350 ppm $CO_2$, 24 °C leaf temperature, 40–60% relative humidity, and 200 µmol $m^{-1}$ flow rate. The leaves of dark-adapted plants were illuminated with red light (600 µmol $m^{-2}$ $s^{-1}$) for 1 h, after which blue light (10 µmol $m^{-2}$ $s^{-1}$) was superimposed on the background red light for 20 min. The induction speeds ($g_{sinduction}$) of red- and blue-light-dependent stomatal opening were evaluated according to the

following equation: $g_{sinduction} = (g_{st} - g_{si}) / (g_{sf} - g_{si})$[69], where $g_{st}$ represents the values of $g_s$ at each time point after red or blue light illumination, $g_{si}$ represents the steady-state values of $g_s$ before red or blue light illumination, and $g_{sf}$ represents the maximum values of $g_s$ under red or blue light. An approximate line was made using $g_{sinduction}$, and the time required for $g_{sinduction}$ to reach 30% of the maximum values was calculated.

For stomatal aperture measurements, the epidermis of dark-adapted plants was incubated in 5 mM MES-bistrispropane (pH 6.5), 50 mM KCl, and 0.1 mM $CaCl_2$ for 2 h in the dark or under red (50 µmol $m^{-2}$ $s^{-1}$) and blue light (10 µmol $m^{-2}$ $s^{-1}$). The stomatal aperture in the abaxial epidermis was observed using an inverted microscope (Eclipse TS100; Nikon) and quantified using the ImageJ 1.48 software (National Institutes of Health).

### Isolation of microsomal fractions from leaves and measurement of ATP hydrolytic activity

Microsomal fractions were isolated from the leaves and ATP hydrolytic activity was measured according to a previously described protocol[70]. The third or fourth leaves were homogenised with a mortar and pestle in an extraction buffer containing 50 mM MOPS-KOH (pH 7.5), 100 mM NaCl, 2.5 mM ethylenediamine-N,N,N′,N′-tetraacetic acid (EDTA), 10 mM NaF, 5 mM dithiothreitol (DTT), 1 mM phenylmethylsulfonyl fluoride (PMSF), and 10 µM leupeptin. After centrifugation at 13,000 $g$ for 10 min at 4 °C, the supernatants were further centrifuged at 100,000 $g$ for 1 h at 4 °C, and the resulting pellet was resuspended in extraction buffer. The microsomal fractions (45 µg of protein, 100 µL) prepared as described above were mixed with an equal volume of ATPase reaction buffer containing 60 mM Tris-MES (pH 6.5), 6 mM $MgSO_4$, 100 mM KCl, 200 mM $KNO_3$, 1 mM ammonium molybdate, 10 µg $mL^{-1}$ oligomycin, 0.1% (v/v) Triton X-100, 0.5 mM PMSF, and 5 µM leupeptin. To determine the ATP hydrolytic activity of the plasma membrane H⁺-ATPase, the mixture was divided in half, and 2 µL of 10 mM sodium orthovanadate, an inhibitor of P-type ATPase, was added to half of the mixture. The reaction was initiated by adding 10 µL of 2 mM ATP to the mixture. The mixture was then maintained at 30 °C for 30 min, and then terminated by adding 1 mL of a stop solution containing 1.3% (w/v) SDS, 0.25% (w/v) sodium molybdate, and 0.15 M $H_2SO_4$. The inorganic phosphate (Pi) released via ATP hydrolysis was determined using a colorimetric method as follows: 50 µL of 1-amino-2-naphthol-4-sulfonic acid (ANSA) solution containing 0.125% (w/v) ANSA, 15% (w/v) $NaHSO_3$, and 1% (w/v) $Na_2SO_4$ were added to the reaction mixture, followed by incubation for 30 min at 24 °C, and the absorption at 750 nm was measured by a spectrophotometer (DU-640, Beckman). The Pi content was determined using a standard curve. The vanadate-sensitive ATP hydrolytic activity was calculated by subtracting the Pi content measured in the presence of sodium orthovanadate from that measured in the absence of sodium orthovanadate and expressed as nmol Pi $h^{-1}$ $mg^{-1}$ protein.

### Statistical analysis

The data analyses carried out in this study were repeated at least three times, and the obtained values are presented as means ± standard deviations (SD). Statistical analysis was performed using the analysis of variance (ANOVA) followed by Tukey's test in Excel 2007 (Microsoft) and Excel Toukei ver. 6.05 and 8.0 (Esumi) and Student's *t*-test in Excel 2021 (Microsoft). *P* value thresholds were *P* < 0.05 or *P* < 0.01.

### Reporting summary

Further information on research design is available in the Nature Portfolio Reporting Summary linked to this article.

## Data availability

The data supporting the findings of this study are provided in the paper, Supplementary Information, and Source Data. Source data are

provided with this paper. The biological materials supporting the findings of this study are available from the corresponding authors upon request. The MS proteomics data have been deposited in ProteomeXchange with the identifier PXD006586, PXD039740, and PXD044476. Source data are provided with this paper.

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

## Acknowledgements

We thank Dr. Asami Hiyama for assistance with phosphoproteomic analysis and Dr. Michito Tsuyama for providing equipment support. This work was supported by JSPS KAKENHI (grant number 21H02511 to A.T., grant number 22K15144 to S.Y., grant number 21H02466 to N.S. and grant number 20H03275 to R.N.), MEXT KAKENHI (grant numbers 21H05665, 22H04726, and 23H04202 to A.T. and grant number 19H05670 to T.K.), the Japan Foundation for Applied Enzymology to A.T., Yamaguchi University Project for Formation of the Core Research Project to A.T., JST SPRING (grant number JPMJSP2111 to S.F.), and Takeda Science Foundation grant to A.T. and R.N.

## Author contributions

S.F., S.Y., T.K., R.N. and A.T. conceived and designed the research; S.F., S.Y., and N.S. performed the research; S.F., S.Y., N.S., K.S., and A.T. analysed the data; and S.F. and A.T. wrote the manuscript with input from all authors.

## Competing interests

The authors declare no competing interests.
