## [Peer Review File · Nature Communications]

Light-induced stomatal opening requires phosphorylation of the C-terminal autoinhibitory domain of plasma membrane H⁺ -ATPaseReviewer #1 (Remarks to the Author):

Fuji et al., studied the signaling of Arabidopsis AHA1 that controls light-induced stomatal opening. The study provided phosphoproteomics analysis revealing some candidate phosphorylation sites in guard cell protoplasts that are induced by blue light. The results are informative resource and of potentially benefit to researches who study plant stomatal signaling. The authors investigated that the phosphorylation of Thr-881 and Thr-948 is crucial for stomatal opening and H⁺-ATPase. I listed some points need to be addressed by the authors.

1. How the mass spectrometry data were processed, how protein phosphorylation is quantified? How the reproducibility of the phosphoproteomics data looks like? As the phosphoproteomic analysis is an important part for this manuscript, LC-MS setting method, the protein database search, data processing... etc. should be presented in the main text. I observed from figure 1 and supplementary file that phosphopeptide amounts are calculated based on integrated peak area value, while such analysis is half-quantification, the most accurate quantitative phosphoproteomics should be performed based on phosphopeptide peak intensity.
2. The authors performed series of physiological experiments to validate the regulatory roles of Thr-881 and Thr-948 of AHA1 in stomatal opening. However, the protein kinases or ligands responsible for the phosphorylation of Thr-881 and Thr-948 are not presented. Considering the importance of these two phosphorylation sites, it would be interesting to screen the upstream kinases for these two key sites.
3. In figure 6, the author states BLUS1 protein kinase mediates the phosphorylation of Thr-881 and Thr-948, however, no experimental evidence provided for this case.
4. The means \pm SD in the analysis of T889 AHA11 in supplementary file figure 1 is missing.

Reviewer #2 (Remarks to the Author):

This paper deals with deeper studies into the phosphorylation of two amino acids, T881 and T948, in the H⁺ATPase AHA1, and the necessity of these two phosphorylation sites to condition stomatal opening in responding to blue and red light.

The stomatal opening responses to blue light and red light are said to be "distinct". Stomatal opening in response to blue light seems better understood molecularly. The blue-light response is enhanced by red light; the stomatal aperture illuminated simultaneously with blue and red light is larger than the sum of the apertures illuminated with blue and red light separately. The general idea is this synergism is due to the interactions between pathways mediated by the blue-light receptors (phototropins) related to those mediated by intercellular CO₂ level (C_i) in leaves through photosynthetic CO₂ assimilation (apparently known since the 1980s according to the listed refereneces). But not mentioned by the authors is the role of C_i seems also controversial because stomata responded to red light even if C_i was artificially maintained constant (Messinger et al. 2006)

Stomatal opening stimulated by blue light is correlated with phosphorylation of two Thr residues in the C-terminal autoinhibitory domain. The phosphorylation status of T948 and T881 are both essential in stomatal response to blue light. But the new data showed that only T881 seems essential in response to red-light, linking it to the photosynthesis pathway.

Two comments.

Authors mentioned that blue light is correlated with the phosphorylation T881 or equivalent residues in AHA1, AHA2, AHA5 and AHA11. And that the phosphorylation of the penultimate T948 or equivalent residues was detected in AHA1, AHA2, AHA4, AHA5, AHA8, AHA11. It is not clear from the information whether the other AHAs were not phosphorylated or simply their protein expression levels were below level of detection.

2. The authors described the blue-light and the red-light effects on stomatal opening are from distinct regulatory pathways. This point is confusing. Indeed, some data in this submission seem to indicate that the light-regulation on the phosphorylation status of T948 is different from that of T881, particularly the differential sensitivity to the electron transport inhibitor DCMU, and to the red/blue light signals. But other lines of evidence suggest that the phosphorylation at the two sites

are connected: the use of phot1 phot2 and the blus1 mutant backgrounds seems to show that the phosphorylation of both sites are affected. Both phosphorylation sites were again affected when PP1s were inhibited by tautomycin. T881 phosphorylation level was lower in the T948A mutant, which led the authors to hypothesize that phosphorylation of T881 is dependent on T948, so again, some functional hierarchy seems to operate here. On the other hand, T881A did not affect the phosphorylation of T948; conversely, T881D did increase the phosphorylation of T948. I have the impression that the two response pathways are intertwined, rather than distinct.

The strong point supporting distinct pathways leading to phosphorylation is the use of DCMU (but has been known for decades). But the diffusible signal(s) from the mesophyll is(are) still not known. There is recent evidence for K^+ as the signal for guard cell non-autonomous stomatal closing, so could this be, in reverse action, the mesophyll signal for non cell autonomous regulation of stomatal opening? There have been suggestion about electrical currents, diffusible hormones, or apoplastic pH. Is there more known on these possibilities? The discussions on these points seem light, in view of their relevance for the T881 phosphorylation evoked by red light.

Thank you for the valuable feedback. We have considered your feedback and made appropriate revisions in the manuscript. Please consider our responses to the comments listed below.

Reviewer #1 (Remarks to the Author):

Fuji et al., studied the signaling of Arabidopsis AHA1 that controls light-induced stomatal opening. The study provided phosphoproteomics analysis revealing some candidate phosphorylation sites in guard cell protoplasts that are induced by blue light. The results are informative resource and of potentially benefit to researches who study plant stomatal signaling. The authors investigated that the phosphorylation of Thr-881 and Thr-948 is crucial for stomatal opening and H⁺-ATPase. I listed some points need to be addressed by the authors.

1. How the mass spectrometry data were processed, how protein phosphorylation is quantified? How the reproducibility of the phosphoproteomics data looks like? As the phosphoproteomic analysis is an important part for this manuscript, LC-MS setting method, the protein database search, data processing... etc. should be presented in the main text. I observed from figure 1 and supplementary file that phosphopeptide amounts are calculated based on integrated peak area value, while such analysis is half-quantification, the most accurate quantitative phosphoproteomics should be performed based on phosphopeptide peak intensity.

Response: Thank you for your valuable insights and suggestions. We have added the details in the methods of LC/MS setting, database search, and data processing to the revised Methods section of the main text.

Reproducibility of the phosphoproteomic workflow was evaluated by analysing five technical replicates of HeLa cell suspensions prior to this study. The average CV of peak area of identified phosphopeptides was 14.6%, and 95% of the peptides had a CV of less than 30%, indicating that the reproducibility of this method is assured. In this study, we performed three independent phosphoproteomic analyses using guard cell protoplasts. Blue light-dependent phosphorylation of Thr-881 or equivalent residues in AHA1, AHA2, AHA5 and AHA11 was detected in all three analyses (Fig. 1b, Supplementary Fig. 1b). Phosphorylation of Thr-948 or equivalent residues in AHA1, AHA2, AHA4, AHA5, AHA8 and AHA11 was detected in all three analyses (Hiyama et al. 2017).

In view of our evaluation 5 technical replicates of HeLa cell suspensions prior to this study, we believe that quantification by integration of XIC peak areas should be more reliable than quantification by peak intensity because more data points can be considered.

(Reference)

Hiyama et al. (2017) *Nat Commun* 8: 1284.

2. The authors performed series of physiological experiments to validate the regulatory roles of Thr-881 and Thr-948 of AHA1 in stomatal opening. However, the protein kinases or ligands responsible for the phosphorylation of Thr-881 and Thr-948 are not presented. Considering the importance of these two phosphorylation sites, it would be interesting to screen the upstream kinases for these two key sites.

Response: We appreciate your feedback and fully agree with your point. We are enthusiastic about identifying the protein kinases responsible for the phosphorylation of the two Thr residues identified in this study. Despite conducting analyses using various methods such as the phosphoproteome, IP-MS, and yeast two-hybrid, we have not yet been able to identify these kinases. However, we remain committed to this pursuit and hope to successfully identify them in future research.

3. In figure 6, the author states BLUS1 protein kinase mediates the phosphorylation of Thr-881 and Thr-948, however, no experimental evidence provided for this case.

Response: Thank you for your insightful comment. We demonstrated that the phosphorylation of Thr-881 and Thr-948 in response to blue light is inhibited in the *blus1* mutant (Fig. 3b), indicating that BLUS1 is involved in mediating the phosphorylation of both Thr residues.

It remains unclear whether the phosphorylation of Thr-881 is caused by a secondary response associated with Thr-948 phosphorylation or is directly regulated by BLUS1-mediated blue light signalling. Our findings indicate that Thr-881 phosphorylation depends on Thr-948 phosphorylation (Fig. 3f, g), suggesting the possibility that blue light signals are transmitted to Thr-948 to facilitate its phosphorylation, subsequently leading to Thr-881 phosphorylation because of conformational changes coupled with Thr-948 phosphorylation without blue light signals. However, besides the requirement of Thr-948 phosphorylation as a prerequisite for Thr-881 phosphorylation, the possibility that Thr-881 may also undergo phosphorylation by protein kinases activated through BLUS1-mediated blue light signalling cannot be ruled out. Thus, we pointed an arrow from BLUS1 to both Thr-881 and Thr-948 in addition to an arrow from Thr-948 phosphorylation to Thr-881 phosphorylation in Fig. 6.

We hope to fully elucidate the mechanisms underlying blue light regulation of Thr-881 phosphorylation in future studies. These potential mechanisms of Thr-881 phosphorylation by blue light have been included in the revised Discussion section.

4. The means \pm SD in the analysis of T889 AHA1 in supplementary file figure 1 is missing.

Response: Thank you for your comment. We provided means \pm SD for the quantification of phosphopeptides containing pThr-889 of AHA11 (Supplementary Fig. 1b).

Reviewer #2 (Remarks to the Author):

This paper deals with deeper studies into the phosphorylation of two amino acids, T881 and T948, in the H⁺-ATPase AHA1, and the necessity of these two phosphorylation sites to condition stomatal opening in responding to blue and red light.

The stomatal opening responses to blue light and red light are said to be “distinct”. Stomatal opening in response to blue light seems better understood molecularly. The blue-light response is enhanced by red light; the stomatal aperture illuminated simultaneously with blue and red light is larger than the sum of the apertures illuminated with blue and red light separately. The general idea is this synergism is due to the interactions between pathways mediated by the blue-light receptors (phototropins) related to those mediated by intercellular CO₂ level (C_i) in leaves through photosynthetic CO₂ assimilation (apparently known since the 1980s according to the listed refereneces). But not mentioned by the authors is the role of C_i seems also controversial because stomata responded to red light even if C_i was artificially maintained constant (Messinger et al. 2006).

Stomatal opening stimulated by blue light is correlated with phosphorylation of two Thr residues in the C-terminal autoinhibitory domain. The phosphorylation status of T948 and T881 are both essential in stomatal response to blue light. But the new data showed that only T881 seems essential in response to red-light, linking it to the photosynthesis pathway.

Two comments.

Authors mentioned that blue light is correlated with the phosphorylation T881 or equivalent residues in AHA1, AHA2, AHA5 and AHA11. And that the phosphorylation of the penultimate T948 or equivalent residues was detected in AHA1, AHA2, AHA4, AHA5, AHA8, AHA11. It is not clear from the information whether the other AHAs were not phosphorylated or simply their protein expression levels were below level of detection.

Response: Thank you for your feedback. In our phosphoproteomic analysis, we did not detect phosphopeptides containing equivalent residues for Thr-881 and Thr-948 in the other AHAs. This could likely be attributed to their low expression levels in guard cells. At the protein expression levels, AHA1 is the major isoform of H⁺-ATPase in guard cells (Yamauchi et al., 2016), and at the mRNA expression levels, *AHA1*, *AHA2*, and *AHA5* are highly expressed in guard cells (Ueno et al., 2005). *AHA3*, *AHA6*, *AHA8*, and *AHA9* are known to be predominantly expressed in pollen and pollen tube (Robertson et al., 2004; Hoffmann et al., 2020). Additionally, *AHA7* and *AHA10* are expressed in roots

and seed coat, respectively (Baxter et al., 2010; Hoffmann et al., 2018).

(References)

- Robertson et al. (2004) *Genetics* 168: 1677–1687.
Ueno et al. (2005) *Plant Cell Physiol* 46: 955–963.
Baxter et al. (2005) *Proc Natl Acad Sci USA* 102: 2649–2654.
Yamauchi et al. (2016) *Plant Physiol* 171: 2731–2743.
Hoffmann et al. (2019) *Physiol Plant* 116: 848–861.
Hoffmann et al. (2020) *Nat Commun* 11: 2395.

2. The authors described the blue-light and the red-light effects on stomatal opening are from distinct regulatory pathways. This point is confusing. Indeed, some data in this submission seem to indicate that the light-regulation on the phosphorylation status of T948 is different from that of T881, particularly the differential sensitivity to the electron transport inhibitor DCMU, and to the red/blue light signals. But other lines of evidence suggest that the phosphorylation at the two sites are connected: the use of *phot1 phot2* and the *blus1* mutant backgrounds seems to show that the phosphorylation of both sites are affected. Both phosphorylation sites were again affected when PP1s were inhibited by tautomycin. T881 phosphorylation level was lower in the T948A mutant, which led the authors to hypothesize that phosphorylation of T881 is dependent on T948, so again, some functional hierarchy seems to operate here. On the other hand, T881A did not affect the phosphorylation of T948; conversely, T881D did increase the phosphorylation of T948. I have the impression that the two response pathways are intertwined, rather than distinct.

The strong point supporting distinct pathways leading to phosphorylation is the use of DCMU (but has been known for decades). But the diffusible signal(s) from the mesophyll is (are) still not known. There is recent evidence for K^+ as the signal for guard cell non-autonomous stomatal closing, so could this be, in reverse action, the mesophyll signal for non cell autonomous regulation of stomatal opening? There have been suggestion about electrical currents, diffusible hormones, or apoplastic pH. Is there more known on these possibilities? The discussions on these points seem light, in view of their relevance for the T881 phosphorylation evoked by red light.

Response: Thank you for your valuable feedback. In this study, we observed different phosphorylation patterns of Thr-881 and Thr-948 in guard cell protoplasts in response to blue and red light. We summarise the key findings obtained from this research.

- 1) Blue light elicited phosphorylation of both Thr-881 and Thr-948, while red light induced phosphorylation of Thr-881 but did not trigger phosphorylation of Thr-948 (Figs. 1a, b and 5a, b).
- 2) Mutants of *phot1 phot2* and *blus1* lost phosphorylation of Thr-881 and Thr-948 in response to

blue light, whereas they showed phosphorylation of Thr-881 in response to red light (Figs. 1a–d, 3b and 5e–h).

- 3) In T948A line, blue light-dependent phosphorylation of Thr-881 was impaired, but red light-induced phosphorylation of Thr-881 was displayed (Fig. 3f, g; Fig. 5i, j). This suggests that phosphorylation of Thr-948 is a prerequisite for blue light-induced phosphorylation of Thr-881, but it is not essential for red light-induced phosphorylation of Thr-881.
- 4) Blue light-induced phosphorylation of Thr-881 and Thr-948 remained unaffected by DCMU, whereas red light-induced phosphorylation of Thr-881 was inhibited by DCMU (Fig. 5c, d and Supplementary Fig. 9) (Suetsugu et al. 2014).

As described above, the phosphorylation of Thr-881 and Thr-948 by blue and red light exhibited differences in reactivity in the mutants, sensitivity to DCMU, and dependence on Thr-948 phosphorylation. Based on these findings, we have reached the conclusion that the phosphorylation of Thr-948 in guard cell protoplasts is regulated by phototropin-mediated blue light signalling, whereas the phosphorylation of Thr-881 in response to blue and red light is regulated by separate signalling pathways originating from phototropin and guard cell photosynthesis, respectively (Fig. 6). We have provided a clearer explanation of these findings in the revised Discussion section. We have also presented additional experimental data that support our interpretation (Fig. 5i, j and Supplementary Fig. 9).

In the guard cell protoplasts used in this study, red light irradiation did not induce phosphorylation of Thr-948 (Fig. 5a, b). In contrast, previous studies using immunohistochemical detection of Thr-948 phosphorylation have shown that red light irradiation of leaves induces phosphorylation of Thr-948 in guard cells, and this phosphorylation is inhibited by DCMU treatment (Ando and Kinoshita, 2018). This discrepancy in the Thr-948 phosphorylation by red light between isolated protoplasts and intact leaves can be attributed to the presence of diffusible signals originating from mesophyll cell photosynthesis (Mott et al., 2008; Fujita et al., 2013).

Among the potential mesophyll signals that induce Thr-948 phosphorylation in guard cells, sucrose produced by mesophyll cell photosynthesis appears to be the most likely candidate. This idea is supported by studies showing that the exogenous application of sucrose to guard cell protoplasts induces phosphorylation of Thr-948 (Okumura et al., 2016). In contrast, a recent investigation has revealed that subjecting leaves to elevated CO₂ concentration results in the dephosphorylation of Thr-948 in guard cells (Ando et al., 2022). This implies that the reduction in intercellular CO₂ concentration within leaves by mesophyll cell photosynthesis could induce phosphorylation of Thr-948 in guard cells. As you suggested, we have included these possibilities in the revised Discussion section.

(References)

Mott et al. (2008) *Plant Cell Environ* 31: 1299–1306.

Fujita et al. (2013) *New Phytol* 199: 395–406.

Suetsugu et al. (2014) *PLoS ONE* 9: e108374.

Okumura et al. (2016) *Plant Physiol* 171: 580–589.

Ando and Kinoshita (2018) *Plant Physiol* 178, 838–849.

Ando et al. (2022) *New Phytol* 236: 2061–2074.

Reviewer #1 (Remarks to the Author):

1. In the 2nd submission, the authors argue that reproducibility of the phosphoproteomic workflow was evaluated by HeLa cell suspensions, I do believe the phosphoproteomic workflow with high reproducibility, while how the global omic dataset or how the reproducibility look like? The authors did not show this, I would suggest submit a PCA plot of volcano plot for this case.
2. As I commented in 1st review, this article focused on the molecular functions of both Thr-881 and Thr-948 of AHA1 in regulating stomatal opening, it would be necessary to screen the upstream kinase or protein interactors for deep understanding of such key phosphorylation sites in regulating light responses. Unfortunately, in the resubmitted version, such information is missing.

Reviewer #3 (Remarks to the Author):

Dear Authors and Editor,

Most seed plants open their pores in the light, to enable the uptake of CO₂ for photosynthesis. This response is provoked by at least two signaling pathways, one that depends on photosynthesis and a second that is initiated by the blue light-sensitive phototropins. The latter response has been studied intensively and has been linked to the activation of plasma membrane H⁺-ATPases in guard cells.

So far, the blue light-induced activation of H⁺-ATPases (mainly AHA1 in guard cells) was explained by the phosphorylation of the penultimate threonine residue (T948) in the C-terminus of this protein. In the manuscript of Fuji et al., the authors explain that it was likely that a second threonine residue (T881) also is involved in regulation of PM H⁺-ATPases, based published work with other cell types as guard cells. They show that this residue indeed gets phosphorylated in guard cells in response to blue light. Based on the results with antibodies that recognize the phosphorylated forms of C-terminus of AHA1, the authors conclude that the phosphorylation of T948 is a prerequisite for phosphorylation of T881, in response to activation of phototropins. However, T881 also get phosphorylated in response to photosynthetic activity in guard cell and in this case the response does not require phosphorylation of T948.

The authors show that the *aha1* mutants, which are complemented with a version of AHA1 that lacks the T at position 881 or T948, do not respond specifically to blue light. Based on these and other results, they conclude that both the phosphorylation of T881 and T948 are required for the blue light-dependent activation of AHA1. However, for the photosynthesis-dependent activation of the H⁺-ATPase, just phosphorylation of T881 seems to be sufficient.

Overall, the data are clearly presented, well documented, and precisely described. The discussion is also precise, but rather long. This section seems to contain some repetitions of observations that are already mentioned in the results, or within the discussion. I only found few points of concern.

1. In Figure 2a it is difficult to see the individual traces. Would it be possible to include a larger version of this graph?
2. Results, lines 187-189. Here it is stated that phosphorylation of T948 is a prerequisite for phosphorylation of T881. Later, it becomes clear that this is only true for the phototropin-dependent response, but not for the photosynthesis-dependent response. It therefore would be better to already point out here at this point in the results that this conclusion is drawn specifically for the blue light response.
3. Legend of Fig. 2b, the authors write "induction speed of stomatal opening" but show the time until 30% of the maximal stomatal conductance has been obtained. This is confusing since faster responses gives lower bars. Please reword the legend.

Reviewer #4 (Remarks to the Author):

The manuscript entitled "Phosphorylation of two Thr residues in the C-terminal autoinhibitory domain of plasma membrane H⁺-ATPase is crucial for light-induced stomatal opening", provides a comprehensive analysis of the phosphorylation events at Thr-881 and Thr-948 in AHA1 in response to blue light, shedding light on their crucial role in stomatal opening and H⁺-ATPase activation. The manuscript establishes an intriguing link between the phosphorylation states of Thr-881 and Thr-948, indicating that Thr-948 phosphorylation is a prerequisite for Thr-881 phosphorylation. The integration of biochemical, genetic, and physiological approaches contributes significantly to our understanding of the complex regulatory mechanisms underlying stomatal responses to light conditions. While the study effectively establishes the correlation between blue light-induced phosphorylation at Thr-881 and Thr-948 and stomatal responses, it falls short in providing mechanistic insights into how this phosphorylation leads to the observed effects. A more in-depth discussion on the signaling pathways and downstream molecular events triggered by this phosphorylation would strengthen the manuscript.

Having carefully reviewed the authors' responses to the initial comments and the revisions made to the manuscript, I have confirmed that the authors have adequately addressed the main points raised during the review process. The revisions have improved the clarity and completeness of the manuscript. Specifically, the authors have provided additional details on the mass spectrometry data processing, quantification methods, and reproducibility of their phosphoproteomic analysis. In addition, the authors have clarified their position on the role of BLUS1 in mediating the phosphorylation of Thr-881 and Thr-948, providing evidence such as the inhibition of phosphorylation in the blus1 mutant. The discussion of differential phosphorylation patterns in response to blue and red light was enriched with additional information to support the conclusion that these phosphorylation events are regulated by distinct signalling pathways originating from phototropin and guard cell photosynthesis. However, the overall impact of the manuscript is significantly diminished by the absence of a more in-depth mechanistic investigation. Nevertheless, based on the authors' thorough responses and the improvements made to the manuscript, I am in favor of publishing the article. I believe that the study makes a valuable contribution to the field and will be of interest to researchers studying plant stomatal signaling.

Reviewer #1 (Remarks to the Author):

1. In the 2nd submission, the authors argue that reproducibility of the phosphoproteomic workflow was evaluated by HeLa cell suspensions, I do believe the phosphoproteomic workflow with high reproducibility, while how the global omic dataset or how the reproducibility look like? The authors did not show this, I would suggest submit a PCA plot of volcano plot for this case.

Response: We appreciate your feedback. As per your request, we have provided the volcano plot of phosphoproteome analysis of *Arabidopsis* guard cells (see below). We confirmed the blue light-dependent phosphorylation in the wild-type and its reduction in the *phot1 phot2* mutant. We are currently submitting another paper using this data, so we would prefer to keep the volcano plot data confidential and not include it in this manuscript.

Phosphoproteome analysis of *Arabidopsis* guard cells.

Volcano plots showing fold change in the levels of identified phosphopeptides in wild-type and *phot1-5 phot2-1* mutant guard cell protoplasts illuminated with blue-light (BL) or red-light (RL). The p-values were calculated using Welch's *t*-test ($n = 3$ biologically independent samples).

2. As I commented in 1st review, this article focused on the molecular functions of both Thr-881 and Thr-948 of AHA1 in regulating stomatal opening, it would be necessary to screen the upstream kinase or protein interactors for deep understanding of such key phosphorylation sites in regulating light responses. Unfortunately, in the resubmitted version, such information is missing.

Response: Thank you for the comment. We have elaborated further on the significance of this point in the Discussion section as follows: "To gain a deeper understanding of the molecular functions of Thr-881 and Thr-948 phosphorylation in light-induced stomatal opening, it is necessary to identify the protein kinases and phosphatases that regulate the phosphorylation at these critical sites in guard cells."

Reviewer #3 (Remarks to the Author):

Dear Authors and Editor,

Most seed plants open their pores in the light, to enable the uptake of CO₂ for photosynthesis. This response is provoked by at least two signaling pathways, one that depends on photosynthesis and a second that is initiated by the blue light-sensitive phototropins. The latter response has been studied intensively and has been linked to the activation of plasma membrane H⁺-ATPases in guard cells.

So far, the blue light-induced activation of H⁺-ATPases (mainly AHA1 in guard cells) was explained by the phosphorylation of the penultimate threonine residue (T948) in the C-terminus of this protein. In the manuscript of Fuji et al., the authors explain that it was likely that a second threonine residue (T881) also is involved in regulation of PM H⁺-ATPases, based published work with other cell types as guard cells. They show that this residue indeed gets phosphorylated in in guard cells in response to blue light. Based on the results with antibodies that recognize the phosphorylated forms of C-terminus of AHA1, the authors conclude that the phosphorylation of T948 is a prerequisite for phosphorylation of T881, in response to activation of phototropins. However, T881 also get phosphorylated in response to photosynthetic activity in guard cell and in this case the response does not require phosphorylation of T948.

The authors show that the *aha1* mutants, which are complemented with a version of AHA1 that lacks the T at position 881 or T948, do not respond specifically to blue light. Based on these and other results, they conclude that both the phosphorylation of T881 and T948 are required for the blue light-dependent activation of AHA1. However, for the photosynthesis-dependent activation of the H⁺-ATPase, just phosphorylation of T881 seems to be sufficient.

Overall, the data are clearly presented, well documented, and precisely described. The discussion is also precise, but rather long. This section seems to contain some repetitions of observations that are already mentioned in the results, or within the discussion. I only found few points of concern.

1. In Figure 2a it is difficult to see the individual traces. Would it be possible to include a larger version of this graph?

Response: Thank you for pointing this out. We have enlarged Fig. 2a as much as possible within the available space.

2. Results, lines 187-189. Here it is stated that phosphorylation of T948 is a prerequisite for phosphorylation of T881. Later, it becomes clear that this is only true for the phototropin-dependent response, but not for the photosynthesis-dependent response. It therefore would be better to already point out here at this point in the results that this conclusion is drawn specifically for the blue light

response.

Response: Thank you for this important comment. We have specified the significance of Thr-948 phosphorylation for blue light-dependent Thr-881 phosphorylation as follows: “These results suggested that Thr-948 phosphorylation was a prerequisite for blue light-dependent Thr-881 phosphorylation.”

3. Legend of Fig. 2b, the authors write “induction speed of stomatal opening” but show the time until 30% of the maximal stomatal conductance has been obtained. This is confusing since faster responses give lower bars. Please reword the legend.

Response: Thank you for the comment. We have revised the legend of Fig. 2b as follows: “Time taken to reach 30% of the maximum values of stomatal conductance in response to RL and BL.”

Reviewer #4 (Remarks to the Author):

The manuscript entitled "Phosphorylation of two Thr residues in the C-terminal autoinhibitory domain of plasma membrane H⁺-ATPase is crucial for light-induced stomatal opening", provides a comprehensive analysis of the phosphorylation events at Thr-881 and Thr-948 in AHA1 in response to blue light, shedding light on their crucial role in stomatal opening and H⁺-ATPase activation. The manuscript establishes an intriguing link between the phosphorylation states of Thr-881 and Thr-948, indicating that Thr-948 phosphorylation is a prerequisite for Thr-881 phosphorylation. The integration of biochemical, genetic, and physiological approaches contributes significantly to our understanding of the complex regulatory mechanisms underlying stomatal responses to light conditions. While the study effectively establishes the correlation between blue light-induced phosphorylation at Thr-881 and Thr-948 and stomatal responses, it falls short in providing mechanistic insights into how this phosphorylation leads to the observed effects. A more in-depth discussion on the signaling pathways and downstream molecular events triggered by this phosphorylation would strengthen the manuscript. Having carefully reviewed the authors' responses to the initial comments and the revisions made to the manuscript, I have confirmed that the authors have adequately addressed the main points raised during the review process. The revisions have improved the clarity and completeness of the manuscript. Specifically, the authors have provided additional details on the mass spectrometry data processing, quantification methods, and reproducibility of their phosphoproteomic analysis. In addition, the authors have clarified their position on the role of BLUS1 in mediating the phosphorylation of Thr-881 and Thr-948, providing evidence such as the inhibition of phosphorylation in the blus1 mutant. The discussion of differential phosphorylation patterns in response to blue and red light was enriched

with additional information to support the conclusion that these phosphorylation events are regulated by distinct signalling pathways originating from phototropin and guard cell photosynthesis. However, the overall impact of the manuscript is significantly diminished by the absence of a more in-depth mechanistic investigation. Nevertheless, based on the authors' thorough responses and the improvements made to the manuscript, I am in favor of publishing the article. I believe that the study makes a valuable contribution to the field and will be of interest to researchers studying plant stomatal signaling.

Response: Thank you for dedicating your time to reviewing our manuscript. We appreciate your thorough review and comments.